

# Beliefs and attitudes towards mental illness: an examination of the sex differences in mental health literacy in a community sample

Raymond J. Gibbons, Einar B. Thorsteinsson and Natasha M. Loi

Department of Psychology, School of Behavioural, Cognitive and Social Sciences, University of New England, NSW, Australia

## ABSTRACT

**Objectives.** The current study investigated mental health literacy in an Australian sample to examine sex differences in the identification of and attitudes towards various aspects of mental illness.

**Method.** An online questionnaire was completed by 373 participants ($M = 34.87$ years). Participants were randomly assigned either a male or female version of a vignette depicting an individual exhibiting the symptoms of one of three types of mental illness (depression, anxiety, or psychosis) and asked to answer questions relating to aspects of mental health literacy.

**Results.** Males exhibited poorer mental health literacy skills compared to females. Males were less likely to correctly identify the type of mental illness, more likely to rate symptoms as less serious, to perceive the individual as having greater personal control over such symptoms, and less likely to endorse the need for treatment for anxiety or psychosis.

**Conclusion.** Generally, the sample was relatively proficient at correctly identifying mental illness but overall males displayed poorer mental health literacy skills than females.

Subjects Epidemiology, Psychiatry and Psychology, Public Health
Keywords Mental health literacy, Sex, Public belief, Mental illness, Vignette

## INTRODUCTION

Mental illness is a predominant issue in public health, contributing to substantial economic and emotional community burden. It is estimated that up to 45% of the Australian population will experience mental illness at some point during their lifetime (*Australian Bureau of Statistics, 2009*). However, not all individuals who experience symptoms of mental illness receive the same level of care or treatment. This is partly attributable to the general public's beliefs and attitudes surrounding mental illness, often referred to as their mental health literacy (e.g., *Jorm et al., 2006*).

The term mental health literacy was first introduced in a study by *Jorm et al. (1997)* investigating public beliefs about the causes and risk factors for depression and schizophrenia. *Jorm et al.* (*1997*, p. 143), described a person's mental health literacy as his or her

Corresponding author
Einar B. Thorsteinsson,
ethorste@une.edu.au,

"knowledge and beliefs about mental disorders that aid the recognition, management or prevention of these disorders." According to *Jorm et al. (1997)*, mental health literacy includes: (a) the ability to recognize and differentiate various types of mental illness and disorders; (b) knowledge of how and where to seek information about risk factors, intervention strategies, and professional help; and (c) attitudes and beliefs that influence a person's ability to identify mental illness and seek appropriate help. Furthermore, an individual's mental health literacy can be influenced by a multitude of factors, including age, remoteness of residency, education, socioeconomic status, and personal experience with mental healthcare (*Dahlberg, Waern & Runeson, 2008*; *Farrer et al., 2008*; *Griffiths, Christensen & Jorm, 2009*; *Kaneko & Motohashi, 2007*).

Studies have shown that the general public historically exhibit poor mental health literacy towards various aspects of mental illnesses (*Goldney, Fisher & Wilson, 2001*; *Jorm et al., 1997*; *Jorm, Christensen & Griffiths, 2005a*; *Jorm, Christensen & Griffiths, 2005b*). For instance, *Jorm et al. (1997)* revealed that only 39% of Australian respondents could accurately recognize symptoms of depression. The beliefs and attitudes of the general public has been shown to be frequently discordant with those held by mental health professionals, with the public frequently viewing medication, hospitalization, and psychiatric treatment as harmful (*Goldney, Fisher & Wilson, 2001*; *Jorm et al., 1997*). A study by *Link et al. (1999)* showed that many symptoms and disorders are not accurately identified by the public as being a mental illness. The results indicated that while there was an overall improvement in mental health literacy, with the public more ably recognizing depression, more positively rating a range of interventions, and holding beliefs and attitudes more consistent with those of mental health professionals, gains still need to be made with respect to schizophrenia and anxiety disorders which are still under-recognized. Interestingly, according to *Andrews* (*1999*, p. 317), both mental health "patients and the media do not distinguish between the non-specific help from counsellors and the specific treatment to be expected from mental health professionals." This inability to discriminate between the types of services offered suggests that the general public's mental health literacy is still largely lacking (*Goldney, Fisher & Wilson, 2001*; *Jorm, Christensen & Griffiths, 2005a*; *Jorm, Christensen & Griffiths, 2005b*; *Jorm & Kelly, 2007*) even for high prevalence disorders such as depression, anxiety, and psychosis. An individual's mental health literacy, including his or her beliefs and attitudes towards mental illness, therefore, may influence or contribute to the formulation of 'lay appraisals.'

Evidence suggests that long before an individual sees a mental health professional, 'lay appraisals' or 'lay diagnoses' are made by individuals, family members, friends, and co-workers regarding the early signs of mental illness (*Hollingshead, 2007*). Given the pervasiveness of these disorders (e.g., *Australian Bureau of Statistics, 2008*; *Sane Australia, 2014*), individuals often have assumptions regarding prognosis as well as opinions relating to the perceived seriousness of the condition, help-seeking or the need for treatment (*Angermeyer, Matschinger & Holzinger, 1998*), and the amount of control they exert over the disorder itself. Lay appraisals are frequently responsible for determining how and when an individual receives treatment for his or her mental illness (*Greenley & Mechanic,*

*1976*; *Pescosolido, Gardner & Lubell, 1998*). As a result, a large portion of people remain undiagnosed and untreated (*Burgess et al., 2009*; *Kessler et al., 1994*).

Some findings suggest that sex differences exist when it comes to public attitudes and beliefs towards mental illness (e.g., *Angermeyer, Matschinger & Holzinger, 1998*; *Cotton et al., 2006*; *Holzinger et al., 2012*; *Jorm et al., 1997*), with *Holzinger et al. (2012)* confirming that females are more likely to advocate professional help than males and female patients are likely to face less societal rejection than male patients. With respect to prevalence of mental illness, females are seen as being at a greater risk of developing mood and anxiety disorders than males (e.g., *Alonso et al., 2004*). Regarding mental health literacy, *Cotton et al. (2006)* investigated young Australians between 12 and 25 years of age. They also revealed that male youths exhibited significantly worse recognition of depressive symptoms than female youths, with 61% of females able to correctly identify depression compared to 35% of males. Furthermore, male youths were less likely than female youths to view seeing a psychologist or counsellor as an appropriate treatment for psychosis. Furthermore, *Holzinger et al.* (*2012*, p. 74) attest that females are more informed about mental illness than males as they have "A stronger tendency to... to conceive problems in psychological terms..."

These studies provide evidence that some sex differences in the mental health literacy of the general public exists. However, it is unknown to what extent such differences currently exists in the Australian general adult population. Expanding our knowledge of sex differences in public awareness of mental illness could help to identify particular areas of mental health literacy in need of improvement specific to each sex. High mental health literacy, including the ability to accurately identify mental illness, may play a pivotal role in help-seeking behaviors and potentially decrease an individual's vulnerability to suicide, particularly for males (*Kaneko & Motohashi, 2007*). Additionally, the identification of such sex disparities in mental health literacy would help to facilitate and guide education programs about mental health, as well as tailor specific individual psycho-education unique to each client.

One of the aims of this study was to build upon the current findings associated with mental health literacy by investigating further the influence that sex has regarding the identification of, and attitudes towards, various aspects of the three major types of mental illness in Australia (i.e., depression, anxiety, and psychosis). As such, the hypotheses are as follows: (1) the general public's mental health literacy is likely to differ significantly for depression, anxiety, and psychosis reflected in an association between identification and type of mental illness, (2) various aspects of the general public's mental health literacy (correct identification) toward depression, anxiety, and psychosis are likely to differ significantly between males and females, and (3) the sex of the individual expressing symptoms of depression, anxiety, and psychosis is likely to influence the general public's mental health literacy. We also examined predictors of (a) the need for treatment, (b) perceived level of control over mental illness, and (c) the perceived susceptibility of each sex to mental illness.
## MATERIALS & METHODS

### Participants

The participants consisted of 381 respondents of varying age, sex, and socio-demographic backgrounds. Eight participants discontinued the questionnaire before completing all of the initial socio-demographic background section and were excluded. The final sample of individuals who participated in this study consisted of 373 participants between the ages of 18 and 84 ($M = 34.87$, $SD = 12.46$) with 28% ($n = 106$) of participants being male ($M = 35.95$, $SD = 12.72$), compared to 72% ($n = 267$) being female ($M = 34.44$, $SD = 12.36$).

Participants were recruited to complete the online questionnaire, created in Qualtrics (www.qualtrics.com), via invitation email through various mailing lists and via messages posted on various social networking sites such as Facebook and Twitter, and by word of mouth. All participants accessed the study via the provided URL. The study was approved by the University of New England's Human Research Ethics Committee (HE11/022).

### Materials and procedures

After reading the online information sheet and consenting to participate in the study, participants completed demographic questions relating to their age, sex, schooling, locality (rural or urban), income, and occupation. Following these, participants read a vignette (random allocation to one of three vignettes with either a male or female protagonist) designed to emulate one of the three most common types of mental illness: major depressive episode ($n = 119$), generalized anxiety disorder ($n = 124$), and psychosis ($n = 130$). Each vignette described an individual who was experiencing symptoms of mental illness at a clinically significant level in which some form of intervention would be recommended as per the criteria stipulated in the *Diagnostic and Statistical Manual of Mental Disorders 5* (*DSM-V*; *American Psychiatric Association, 2013*). The depression and psychosis vignettes were adapted from the depression and schizophrenia vignettes used by *Jorm et al. (1997)*.

The names 'John' and 'Jane' were chosen for use in the vignettes due to the fact that they are common Australian names with no cultural association to any minority groups. The age of the protagonist was kept consistent at 30 years of age in order to avoid the potential confound of developmental and psychosocial difficulties that often occur in childhood and adolescence as well as to avoid age-specific neurological and physical conditions often present in older adults. Participants were initially asked, in an open-ended question, to identify the disorder presented in the vignette. (i.e., "In five words or less, what would you say, if anything, is wrong with the individual in the vignette?"). Additional forced choice questions, rated from 0 to 6, were designed to ascertain: (a) participants' perceptions of the seriousness of the symptoms described (i.e., "To what extent would you rate the problems of the individual in the story as being serious?"); (b) the likelihood that treatment might be required (i.e., "To what extent would some form of treatment or intervention be required for the individual in the story?"); and (c) the perceived level of control the individual in

the vignette has over the symptoms described (i.e., "To what extent are the problems of the individual in the story within his or her control?").

The final questions related to participants' sex perceptions regarding mental illness (e.g., "Which group of people would you consider to be most likely to experience problems similar to those of the individual in the story?"). Participants were required to respond by choosing whether males, females, or both were equally likely to experience problems.

The following overall guidelines were established for the purposes of providing consistency and accuracy in distinguishing between correct and incorrect responses to the above questions: (a) the use of the general category of mental illness (i.e., depression, anxiety, and psychosis), or a derivative of the word (e.g., depressive, depressed, anxious, or psychotic) was regarded as a correct response; (b) the use of the exact *DSM-V* diagnostic criteria, or subtype there of (e.g., major depression, dysthymia, generalized anxiety disorder, or paranoid schizophrenia) was also regarded as a correct response; (c) references to symptoms of a mental illness rather than the illness itself were not regarded as correct responses; (d) the term "stressed" was not accepted as a correct identification of anxiety on the basis that it is frequently used as a colloquial term that can encompass a broad range of symptoms, some of which are often not associated with anxiety. Similarly, the terms "paranoid" and "delusional" were not accepted as correct identification of psychosis as they may also be used in colloquial contexts, and refer to the symptoms of schizophrenia rather than the illness itself; (e) misspelt words or phrases were accepted as correct providing it was discernible as to what mental illness was intended; and (f) accurate responses were maintained as correct regardless of additional information or diagnoses that were provided beyond that of the correct diagnosis, as the participant demonstrated the ability to identify the mental illness in question.

The study took approximately 15 min to complete.

## Statistical analysis

All statistical analyses were conducted using SPSS Version 21. Chi-square was utilized to examine associations between nominal variables including identification of type of illness, participant and protagonist sex, and susceptibility of mental illness. Analysis of variance (ANOVA) was used to examine group differences in the outcome (interval level) variables including perceived seriousness, need for treatment, and personal control.

## RESULTS

### Identification of mental illness

Hypothesis 1 related results that showed there was a strong association between correct identification and type of mental illness, $\chi^2(2, N = 373) = 52.11$, $p < .001$, $\phi = .37$, with 86% of participants correctly identifying depression, 57% of participants correctly identifying anxiety, and 42% correctly identifying schizophrenia. Testing Hypothesis 2 showed that there was also a weak but statistically significant association between correct identification and sex, $\chi^2(1, N = 373) = 4.70$, $p = .03$, $\phi = -.11$, with 52% of male and 64% of female participants correctly identifying the mental illness. Examining

**Table 1** Between Groups ANOVAs for the effects of illness, sex, and protagonist sex on perceived seriousness, need for treatment, and personal control.

| Measures | Perceived seriousness | | Need for treatment | | Personal control | |
|---|---|---|---|---|---|---|
| | $F$ | Partial $\eta^2$ | $F$ | Partial $\eta^2$ | $F$ | Partial $\eta^2$ |
| Illness type (I) | 69.74[***] | .28 | 49.81[***] | .22 | 20.50[***] | .10 |
| Sex (S) | 6.19[*] | .02 | 18.29[***] | .05 | 7.01[**] | .02 |
| Protagonist sex (PS) | 4.73[*] | .01 | 1.04 | <.01 | 0.79 | <.01 |
| I × S | 2.01 | .01 | 3.11[*] | .02 | 0.36 | <.01 |
| I × PS | 0.02 | <.01 | 0.16 | <.01 | 0.30 | <.01 |
| S × PS | 2.22 | .01 | 0.006 | <.01 | 4.70[*] | .01 |
| I × S × PS | 0.76 | <.01 | 0.44 | <.01 | 1.42 | .01 |

**Notes.**
[*] $p < .05$.
[**] $p < .01$.
[***] $p < .001$.

**Table 2** Means and standard deviations for condition, sex of participants and protagonist by the perceived seriousness, need for treatment, and personal control.

| Measure | Perceived seriousness | Need for treatment | Personal control |
|---|---|---|---|
| **Illness** | | | |
| Depression | $5.57 (1.04)_a$ | $5.13 (1.10)_a$ | $3.81 (1.37)_a$ |
| Anxiety | $4.87 (1.18)_b$ | $4.67 (1.21)_b$ | $4.04 (1.48)_a$ |
| Psychosis | $6.46 (0.75)_c$ | $6.09 (0.90)_c$ | $2.86 (1.56)_b$ |
| **Participant** | | | |
| Male | $5.46 (1.30)_a$ | $4.94 (1.26)_a$ | $3.87 (1.49)_a$ |
| Female | $5.72 (1.14)_b$ | $5.46 (1.19)_b$ | $3.43 (1.57)_b$ |
| **Protagonist** | | | |
| Male | $5.74 (1.20)_a$ | $5.36 (1.21)_a$ | $3.56 (1.64)_a$ |
| Female | $5.57 (1.19)_b$ | $5.26 (1.25)_a$ | $3.55 (1.48)_a$ |

**Notes.**
Values within variables in columns that share a subscript are not different by alpha criterion of .05 (Sidak adjusted).

Hypothesis 3 showed that there was only a weak association between correct identification and protagonist sex, $\chi^2(1, N = 373) = 0.33, p = .056, \phi = .03$, with 62% of male and 59% of female protagonist mental illness correctly identified.

## The need for treatment

Four three-way between groups ANOVAs, type of illness by sex by protagonist sex, were conducted for perceived seriousness, need for treatment, and personal control, see Table 1. The means and standard deviations are reported in Table 2 along with any post hoc analysis of main effects.

The interaction effect between mental illness and participant sex was significant for need for treatment, indicating that the effect that participant sex has on the perceived

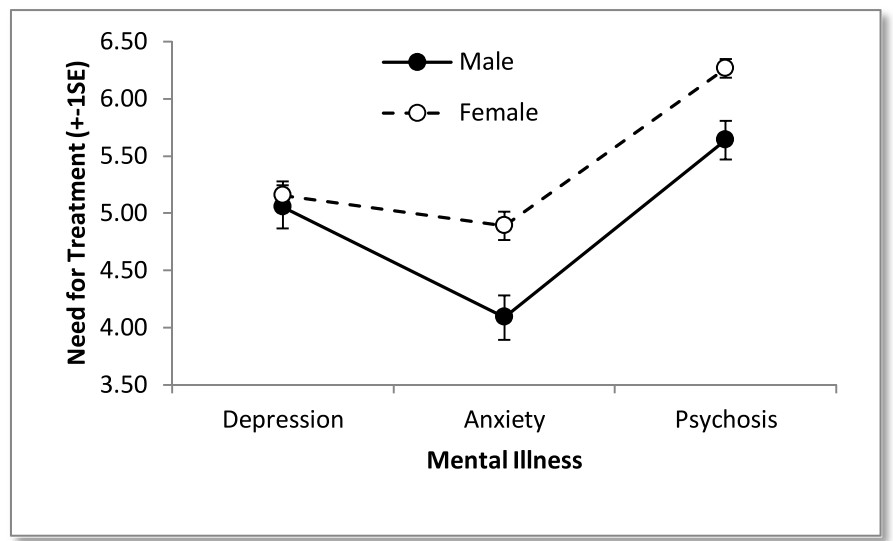

**Figure 1** Image of need for treatment expressed by male and female participants towards the three types of mental illness.

need for treatment for mental illness is dependent on the type of mental illness, see Fig. 1. Analysis of simple effects was conducted using separate one-way between groups ANOVAs to determine the independent effects of participant sex and type of mental illness. For depression there was no significant difference between participant sex, $F(1, 117) = 0.21$, $p = .65$, partial $\eta^2 < .01$. For anxiety there was a significant difference between participant sex, $F(1, 122) = 11.80$, $p = .001$, partial $\eta^2 = .09$, with females ($M = 4.89$, $SD = 1.17$) perceiving a higher need for treatment than males ($M = 4.09$, $SD = 1.14$). For psychosis there was a significant difference between participant sex, $F(1, 128) = 13.84$, $p < .001$, partial $\eta^2 = .10$, with females ($M = 6.27$, $SD = 0.79$) perceiving a higher need for treatment than males ($M = 5.64$, $SD = 1.02$).

### Perceived level of control

There was a significant interaction between participant and protagonist sex for personal control, see Table 1, indicating that the effect that participant sex has on perceived level of control over mental illness is dependent on the type of protagonist sex. Analysis of simple effects was conducted using separate one-way between groups ANOVAs to determine the effects of participant and protagonist sex. For males there was no significant difference between protagonist sex, $F(1, 104) = 3.05$, $p = .08$, partial $\eta^2 = .03$. For females there was no significant difference between the protagonist sex, $F(1, 265) = 0.74$, $p = .39$, partial $\eta^2 < .01$. For male protagonists there was no significant difference between participant sex, $F(1, 180) = 0.23$, $p = .64$, partial $\eta^2 < .01$. For female protagonists there was a significant difference between participant sex, $F(1, 189) = 10.77$, $p = .001$, partial $\eta^2 = .05$, with male participants rating a higher level of perceived control than females, see Fig. 2.

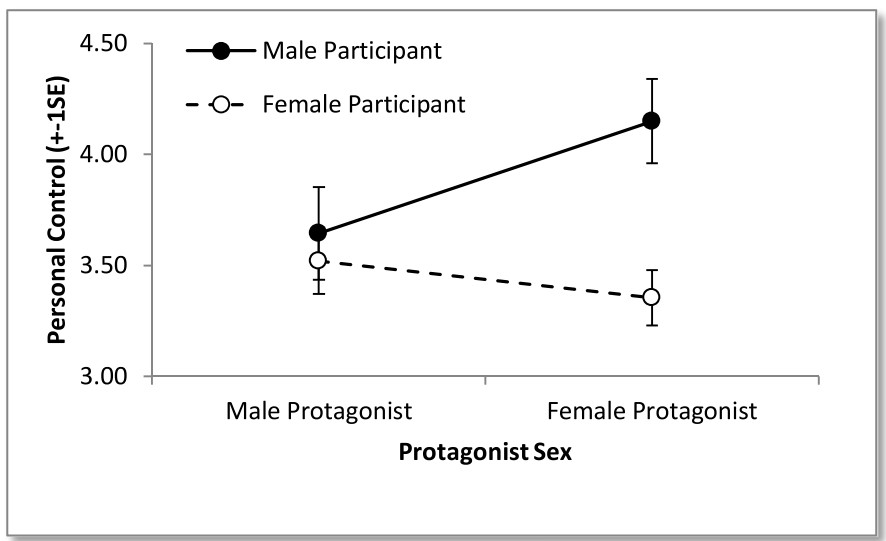

**Figure 2** Image of perceived level of personal control over mental illness for each protagonist sex as rated by each participant sex.

## Perceived sex susceptibility

Three chi-square tests were performed to test if there was a relationship between perceived sex susceptibility and the type of mental illness, participant sex, and protagonist sex. There was a significant association between perceived sex susceptibility and type of mental illness, $\chi^2(4, N = 373) = 22.48$, $p < .001$, $\phi = .25$. From the participants who received the depression vignette, 13.8% indicated that males are more susceptible, 15% indicated that females are more susceptible, and 72% indicated that males and females are equally susceptible. From the participants who received the anxiety vignette, 13% indicated that males are more susceptible, 23% indicated that females are more susceptible, and 63% indicated that males and females are equally susceptible. From the participants who received the psychosis vignette, 28% indicated that males are more susceptible, 6% indicated that females are more susceptible, and 66% indicated that males and females are equally susceptible.

There was a significant association between perceived sex susceptibility and participant sex, $\chi^2(2, N = 373) = 11.00$, $p = .004$, $\phi = .17$. From the male participants, 29% indicated that males are more susceptible, 9% indicated that females are more susceptible, and 62% indicated that males and females are equally susceptible. From the female participants, 15% indicated that males are more susceptible, 17% indicated that females are more susceptible, and 68% indicated that males and females are equally susceptible.

## Post hoc analysis: sex susceptibility

For methodological reasons, we examined whether participants would regard the sex of the protagonist as important to their susceptibility to mental illness. There was a significant association between perceived sex susceptibility and protagonist sex, $\chi^2(2, N = 373) = 35.70$, $p < .001$, $\phi = .31$. From the participants who received a male protagonist vignette, 29%

indicated that males are more susceptible, 6% indicated that females are more susceptible, and 65% indicated that males and females are equally susceptible. From the participants who received a female protagonist vignette, 9% indicated that males are more susceptible, 22% indicated that females are more susceptible, and 69% indicated that males and females are equally susceptible.

## DISCUSSION

This study explored whether the mental health literacy of members of the general public was influenced by (a) the type of mental illness, (b) the sex of the individual experiencing the symptoms of mental illness, and (c) the sex of the individual identifying and appraising the mental illness. Participants read a vignette describing either a male or female experiencing clinically significant symptoms of depression, anxiety, or psychosis. Participants were then asked a number of questions in order to assess various aspects of their mental health literacy. The results were consistent with the hypothesis that some aspects of mental health literacy are influenced by the type of mental illness, sex of the participant, and sex of the protagonist.

As hypothesized, the type of mental illness directly influenced participants' abilities to accurately identify the presenting symptoms, with depression resulting as the most readily identifiable mental illness (85.7%), followed by anxiety (56.5%), and then psychosis (41.5%). These findings suggest that the general public is relatively good at identifying symptoms of depression, but find it significantly more difficult to identify anxiety and psychosis. This is consistent with previous research that has shown depression to be more readily identifiable by the general public than psychosis (e.g., *Reavley & Jorm, 2011*). A number of factors may contribute to the increased recognition of depression over anxiety and psychosis. First, a greater prevalence of depression and anxiety in Australia compared to psychosis may result in members of the general public having greater exposure and experience with such mental illnesses. Second, given the exposure provided by organizations such as beyondblue, the term *depression* is more readily used in the Australian vernacular and is associated with milder social stigmas (*Jorm, Christensen & Griffiths, 2005b*). Finally, compared to depression and anxiety, which refer to broad categories of psychological symptoms, psychosis refers to a more specific set of psychological symptoms outlined in the *DSM-V*, subsequently making it more difficult to accurately identify (*American Psychiatric Association, 2013*; *Australian Bureau of Statistics, 2009*).

Across the measures of level of seriousness, participants showed the greatest level of concern towards individuals with psychosis, followed by depression, and anxiety. This is consistent with evidence that has shown that psychosis and depression are associated with an increased level of morbidity and mortality, when compared to the general population, and in particular, as relates to serious cardiovascular events (*Casey et al., 2004*; *Musselman, Evans & Nemeroff, 1998*; *Wulsin, Vaillant & Wells, 1999*). Similarly, psychosis and depression are associated with increased levels of suicide attempts compared to the general population. In fact, suicide is the leading cause of premature death among individuals with schizophrenia (*Fenton, 2010*). These findings suggest that the general

public have accurate perceptions of the seriousness of mental illnesses and support the finding by *Reavley & Jorm (2012)* which identified a growing trend of improved mental health literacy of the general public over the past decade. As hypothesized, a significant difference existed between each participant sex and their respective ability to accurately identify the presenting symptoms, with 64% of females able to correctly identify the mental illness provided compared to 52% of males. Across the measures of level of seriousness, females displayed an overall tendency to perceive the symptoms of mental illness as more serious compared to males. These findings are congruent with previous studies suggesting that when compared with females, males display a poorer ability to correctly identify mental illness as well as more definitive (if imprecise) attitudes towards the various aspects of mental illness (*Cotton et al., 2006*; *Kaneko & Motohashi, 2007*). For example, in *Cotton et al.*'s study, males believed that the prevalence of mental illness was only 1% and that treatment by a medical professional was unlikely for psychosis. One possible explanation for these findings is that females may be inherently more psychologically minded, introspective, and emotionally aware, thus increasing the likelihood that they would (a) engage in conversations relating to emotional and psychological difficulties, (b) have contact or interactions with individuals who have a mental illness, and/or (c) participate in studies relating to mental health literacy (*Petrides, Furnham & Martin, 2004*).

Finally, as hypothesized, the sex of the protagonist presented in the vignette provided had an influence on a number of aspects of mental health literacy including the perceived level of seriousness, degree of personal control, and sex susceptibility. Individuals who received a vignette with a male protagonist displayed a tendency to report marginally higher levels of perceived seriousness compared to individuals who read about a female protagonist. Furthermore, of the individuals provided with a male protagonist vignette, 29% indicated that males were more likely to be susceptible to mental illness compared to 6% of individuals who indicated that females were more likely. Conversely, when presented with a female protagonist vignette the opposite trend was observed, with 22% indicating that females were more likely to be susceptible to mental illness compared to 9% of individuals who indicate that males were more likely. However, it is worth noting that the majority of participants considered both males and females as equally susceptible. Considering need for treatment, results indicated that participant, and not protagonist, sex was the principal factor. This was true for anxiety and psychosis only, with females more likely than males to indicate that treatment was required for these conditions. This finding is consistent with previous research (e.g., *Holzinger et al., 2012*) that also indicates that females are more likely to endorse professional help for mental illness than males. Further findings indicated that when provided with a female protagonist vignette, males displayed a tendency to perceive a higher level of personal control over mental illness than females. However, when provided with a male protagonist vignette no sex differences were observed. These results are consistent with, and build upon, preliminary evidence obtained by *Jorm et al. (1997)* by suggesting that protagonist sex may influence mental health literacy not only through self-reported beliefs, but also as a result of innate perceptions regarding mental illness.

## Limitations

Similar to other research conducted regarding mental health literacy, this research adopted the use of a brief written vignette. The use of such a vignette presents two distinct complications. First, it is uncertain whether a written description of symptoms of mental illness elicits the same attitudes and perceptions as obtaining the same information from face-to-face observations and verbal communication. It is likely that overt body language and non-verbal communication (i.e., facial expressions, eye contact, tone of voice) would provide additional information from which to identify and evaluate an individual's symptoms of mental illness.

Second, it is unknown whether the level of concern expressed towards a vignette of mental illness is comparable or equivalent to that expressed towards a real person and whether there are any differences in the subsequent therapeutic actions taken from such concern (i.e., help seeking behavior).

About 72% of the participants in the present study were females. Such a discrepancy may have potentially created an overrepresentation of female beliefs and attitudes when performing comparisons based on the entire sample. However, there were no statistically significant sex-related interactions except for need for treatment. Due to a limited sample size and the scope of the initial aims in the present study, we were not able to control for various demographics. However, the data set is available (*Gibbons, Thorsteinsson & Loi, 2015*) and can be used in conjunction with similar information to enhance power in future analyses.

The vignettes employed in this study depicted three different mental illnesses clearly distinguishable from each other through the absence of overlapping symptoms or comorbid diagnoses involving substance abuse, medical conditions, trauma, personality disorders, or intellectual difficulties. Comorbidity among mental illness is extremely high and would inevitably have a large impact on an individual's ability to accurately identify and evaluate mental illness. As such, the findings from this study may not be extrapolated to apply to common situations whereby comorbidity is present (*Kessler et al., 2005*). Similarly, the vignettes utilized in this study described the protagonists as being 30 years of age in order to maintain consistency. Unique age-related difficulties in younger (e.g., puberty) and older populations (e.g., cognitive decline) may potentially complicate the identification of mental illness (*Bartels, 2004*; *Pottick et al., 1995*).

Future research on mental health literacy may consider several issues: (a) considering whether the trends displayed using vignettes are comparable to 'real life' symptoms, (b) the extent to which expressed concerns towards mental illness equate to therapeutic action, (c) the influence of comorbidity towards mental health literacy, (d) the influence of the age of the protagonist, (e) the effect of utilizing a vignette describing sub-clinical everyday problems as a comparison, (f) exploration of the possible reasons behind participants' mental health literacy, and (g) examining comparisons between the mental health literacy of adolescents and adults. While studies comparing the mental health literacy of different age groups within adolescent and adult populations (e.g., *Farrer et al., 2008*; *Jorm, Morgan & Wright, 2008*), no research has compared adolescents and adults within the same study.

## Conclusions

The findings from this study suggest that the Australian general public is relatively proficient at correctly identifying mental illness, in particular symptoms of depression. The general public also displayed relatively accurate perceptions of the severity and seriousness of symptoms of depression, anxiety, and psychosis. Males exhibited poorer mental health literacy skills than females, with males being less likely to correctly identify the type of mental illness, displaying a tendency to rate symptoms as being less serious, and perceiving greater personal control over mental illness. These findings help to identify the area of public mental health literacy that may be improved in the Australian general public, namely education programs targeted towards increasing awareness of mental illness in the male population. Increased awareness of how sex may potentially influence people's ability to identify mental illness and subsequently evaluate its severity may help to guide mental health professionals in clinical decision making. In addition, this awareness may assist the general public make more accurate 'lay appraisals' of mental illness; however, further research is needed in order to determine whether the public's attitudes and beliefs towards mental health literacy derived from vignette research is equivocal to that derived from real life face-to-face consultations and how such beliefs and attitudes impact on actual help-seeking behaviors.

### Funding

The authors declare there was no funding for this work.

### Competing Interests

The authors declare there are no competing interests.

### Author Contributions

- Raymond J. Gibbons conceived and designed the experiments, performed the experiments, analyzed the data, contributed reagents/materials/analysis tools, wrote the paper, prepared figures and/or tables, reviewed drafts of the paper.
- Einar B. Thorsteinsson conceived and designed the experiments, analyzed the data, contributed reagents/materials/analysis tools, wrote the paper, reviewed drafts of the paper.
- Natasha M. Loi wrote the paper, reviewed drafts of the paper.

### Human Ethics

The following information was supplied relating to ethical approvals (i.e., approving body and any reference numbers):

University of New England Human Research Ethics Committee, HE11/022.

## Data Deposition

The following information was supplied regarding the deposition of related data:
FigShare: http://dx.doi.org/10.6084/m9.figshare.1392513.

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
