# Peer review of "Beliefs and attitudes towards mental illness: an examination of the sex differences in mental health literacy in a community sample"

_PeerJ, doi:10.7717/peerj.1004_

## Round 0.1 · original submission · Major Revisions

Could you please consider each of the recommendations made by the two reviewers. Either make the revision they suggest or provide me with a reason for not making it in your response letter.

Reviewer 1 ·

Basic reporting

The paper is structured appropriately, clearly written for the most part, and offers a readable summary of the concept of mental health literacy and the findings in the area to date in the Introduction. However, often the references in the Introduction do not cite the most up-to-date publications. For example, the reference to Shapiro, Skinner and Kessler (1984) on line 63 could be replaced with a paper by Burgess and colleagues which makes the same point, but is specific to the Australian population, and was published more recently (Burgess, P. M., Pirkis, J. E., Slade, T. N., Johnston, A. K., Meadows, G. N., & Gunn, J. M. (2009). Service use for mental health problems: Findings from the 2007 National Survey of Mental Health and Wellbeing. Australian & New Zealand Journal of Psychiatry, 43(7), 615-623. doi: 10.1080/00048670902970858).

Lines 65-71 have no reference, but seem to refer to either one of these papers:

Reavley, N. J., & Jorm, A. F. (2011). Recognition of mental disorders and beliefs about treatment and outcome: Findings from an Australian National Survey of Mental Health Literacy and Stigma. Australian & New Zealand Journal of Psychiatry, 45(11), 947-956. doi: 10.3109/00048674.2011.621060
or
Reavley, N. J., & Jorm, A. F. (2012). Public recognition of mental disorders and beliefs about treatment: Changes in Australia over 16 years. British Journal of Psychiatry, 200, 419-425. doi: 10.1192/bjp.bp.111.104208

These papers could also be used to update the references in lines 228-9 and 245.

While the history of mental health literacy is clearly explained in the Introduction, the review of sex differences in mental health literacy is more vague. The paragraph beginning on line 72 seems to be quite important to the paper’s premise, but is disjointed and seems to skim over previously found sex differences in other studies. It may be worthwhile expanding on the findings of the Holzinger et al (2012) review, as it seems to pertain more to adult populations and discusses results that are relevant to the attitudes asked about in this study (e.g. reactions to people with mental illnesses). The importance of the study and the hypotheses are well written, although the sentence “High mental health literacy…[may] potentially decrease an individual’s vulnerability to suicide” needs a reference.

The Results and Discussion sections are well summarised. In the Results section, however, Figures 3, 4 and 5 seem unnecessary, as these results are reported in-text as percentage values (lines 194-213); readers do not need to see the number of participants who gave each answer if they already know the percentage values.

The reference on line 404 of the References section is not mentioned in the text of the paper.

Experimental design

The research question and hypotheses are well defined. However, the Materials and Methods section requires substantial clarification:
• The paragraph beginning on line 107 explains how participants were recruited but not how they received the survey. Was the survey linked into the advertisements or were participants emailed the link after expressing their interest, for example?
• In the Materials and Procedures section:
o How many participants received each vignette?
o What type of anxiety disorder was presented in the anxiety vignette?
o Was the sex of the character in the vignette also randomised?
o How long did the survey take participants to complete?
o The survey questions need to be described in more detail (lines 127-137). Lines 127-8 state that “a series of questions…were created to ascertain various aspects of their mental health literacy...(i.e., “In five words or less, what would you say, if anything, is wrong with the individual in the vignette?”) Was this the only question they were asked? If so, the sentence might read better as “Participants were first asked to identify the disorder presented in the vignette. Additional questions were designed to ascertain…” Each of the questions which have results mentioned in the Results section need to be described in the Methods section; the questions relating to the vignette character’s need for treatment or risk of self-harm are not mentioned at all, which makes it difficult for the reader to understand the references to these variables in Tables 1 and 2.
o The format of each question should be outlined as well, e.g. participants were asked to rate the seriousness of the problem (lines 131-2), but the rating scale used is not described. Similarly, lines 135-7 do not mention whether the questions relating to sex perceptions were forced choice, open ended, rated on a scale, or otherwise; Figures 3, 4 and 5 imply that the questions were forced choice, but this should be clarified.
o Does the paragraph describing correct and incorrect responses (beginning on line 138) relate only to the recognition of disorder question, or did these criteria apply to other questions in the survey? What were the criteria used to determine accurate perceptions of sex susceptibility?

Validity of the findings

The Discussion section could benefit from additional references which support the points made by the authors when explaining their results, e.g. line 232-3 “depression is…associated with fewer and milder social stigmas” is not referenced. The following reference may also be of use in understanding why depression is better recognised than other mental illnesses: Jorm, A. F., Christensen, H., & Griffiths, K. M. (2005). The impact of beyondblue: the national depression initiative on the Australian public's recognition of depression and beliefs about treatments. Australian & New Zealand Journal of Psychiatry, 39(4), 248-254. The paragraphs beginning on lines 248 and 253 could be joined, as they seem to discuss the same finding. The Discussion would also benefit from a clearer explanation of what this study adds to the literature on sex differences in mental health literacy in the Australian public and a brief comment on the strengths of the study. The concluding sentence (lines 322-325) introduces new information into the manuscript; it might instead be better placed earlier in the Discussion, with some comments on how the results could inform research and practice, e.g. should education programs targeting mental health literacy in males focus more strongly on information about a specific disorder, how to help someone correctly identify a mental illness, or on reducing the stigma associated with the belief that people with mental illnesses lack personal control?

In the limitations section, the papers referred to as the Jorm et al and Link et al citations on lines 295-96 do not show that males are less likely than females to participate in mental health literacy research; both studies involved nationally representative samples, so rates of participation would have been equivalent to population demographics, and neither discuss how often males refused to participate compared to females. On lines 313-4, “exploration of the possible reasons behind participants’ mental health literacy” requires further explanation; it is not clear what this means. Similarly, the difference between “youths” and “adolescents” is unclear on line 315. It may be worthwhile mentioning that while there are published studies that compare the mental health literacy of different age groups within the adult and adolescent categories (e.g. Farrer, L., Leach, L., Griffiths, K. M., Christensen, H., & Jorm, A. F. (2008). Age differences in mental health literacy. BMC Public Health, 8(125). doi: 10.1186/1471-2458-8-125; Yap, M. B. H., Wright, A., & Jorm, A. F. (2011). First aid actions taken by young people for mental health problems in a close friend or family member: Findings from an Australian national survey of youth. Psychiatry Research, 188, 123-128. doi: 10.1016/j.psychres.2011.01.014; Jorm, A. F., Morgan, A. J., & Wright, A. (2008). First aid strategies that are helpful to young people developing a mental disorder: Beliefs of health professionals compared to young people and parents. BMC Psychiatry, 8, 42. doi: 10.1186/1471-244X-8-42), there has not yet been a study that compares adolescents and adults within the same survey, although often Australian mental health literacy surveys for youth and adults use similar questions, and so can be compared (cf the Jorm, Morgan and Wright reference above).

Additional comments

It would be helpful to the reader if the expression was tidied to improve clarity, e.g. instead of “a male protagonist vignette” (lines 263, 265-266, and elsewhere in the Discussion), try “a vignette with a male protagonist” or “the John vignette”; in line 93, “the hypotheses proposed to be explored by the present research are as follows” could simply read “the study’s hypotheses are as follows”; the concluding sentence (lines 322-325) states that there are “numerous areas of public mental health literacy that may be improved,” but only offers one example of this. The italics on lines 169, 170 and 180 are inconsistent with the rest of the variables mentioned in the manuscript.

·

Basic reporting

Title and Abstract
1) Consider rephrasing the title to capture what was examined in the study in light of the hypotheses. At the moment the title seems to be only relevant to the examination done in relation to the second hypothesis.
2) Also include findings relevant to the other hypotheses in the abstract.
3) When considering the second hypothesis, it would seem that the examination of sex differences in mental health literacy was being done separately for the three disorders. Hence, it would seem more relevant to report findings in relation to such an examination (i.e., findings in relation to perceptions of need for treatment among males and females for the three disorders) than the overall gender differences observed.

Introduction
4) I feel that the flow and structure of the introduction could be improved to provide a stronger rationale for the hypotheses.
- For example, findings regarding recognition of disorders are presented at two points in the introduction (lines 41-44 and 63-71). Consider combining these sections into one paragraph.
- As the second and third hypotheses are examining two different aspects it might be better to discuss research relevant to each of these separately instead of together (lines 72-80). It would also be helpful if there is more review of research relevant to each of these.
- As there are various aspects of mental health literacy that could be examined, ensure that this section provides a rationale for the elements that were examined. I assume the discussion of ‘lay appraisals’ has been provided in relation to aspects such as perceptions of severity of illness, personal control etc. However, a rationale for the examination of these specific aspects has not been provided.
- Some of the literature discussed appears no quite relevant to what was examined in the study. Hence, consider replacing these sections with a review of literature relevant to the study (e.g., lines 47-55).

Minor revisions relating to editing
• In the abstract, specify that it is the mean age that is reported in the results section and that it is for the entire sample.
• Line 31- check if the page number of the citation is accurate
• Lines 48-49- would it be possible to paraphrase instead of providing a quotation? (if retaining related section)
• Line 50- “lends further weight to the research” the study has still not been described. Therefore, it is not possible to refer to it as yet.
• Line 115-116- What is meant by “three most common types of symptoms associated with mental illness”? Clarify.
• 238-242- the idea presented here is not clear
• 253-255- rephrase sentence. Clarify what is meant by restrictive here.

Experimental design

Introduction
1) Line 90- there is reference to one of the aims of the study. Were there other aims? Furthermore, if this was the only aim of the study, the first hypothesis is not relevant to this aim.
2) As mentioned in the previous section, when reading the second hypothesis it seems that examination of sex differences in mental health literacy was being done separately for the three disorders. However, some of the analyses do not reflect this. Consider whether such analyses would be suitable (in light of the low number of male participants and issues relating to the power of the analyses) or whether it is necessary to re-formulate this hypothesis. This issue is relevant for the third hypothesis as well.
3) Consider if it would be better to reformulate the hypotheses as aims. The hypotheses lack specificity when there is use of terms such as “various aspects”. Furthermore, as in fact there are various aspects examined in this study, with findings for each of these differing, it is difficult to state whether the hypotheses were supported or not.

Methods
4) There is no mention that the cases of ‘John’ and “Jane’ were also randomly assigned. Please mention this.
5) Please describe the rating scales that were used for assessing perceptions about seriousness and personal control relating to the illness.
6) There is no mention of examining perceptions about the need for treatment and risk of self-harm. Describe how these were examined.
7) Also provide the response options for the sex susceptibility question.
8) Include a section on the statistical analyses that were undertaken.
9) There is mention of obtaining information about whether participants were from rural or urban areas, their occupation, schooling etc. Were these factors controlled for in the analysis? Comment on how these could have affected the findings.

Validity of the findings

Results
1) Although results relating to recognition of and sex susceptibility to the different disorders (relevant to first hypothesis) are provided, there is no mention of results in relation to the other aspects that were examined. Furthermore, consider incorporation of a discussion of these findings in the discussion section, e.g., differences in perception for the need for treatment
2) 180-189- Is this analysis relevant to any of the hypotheses? Or would this be relevant to yet another hypothesis?
3) Lines 191-201- consider presenting the Chi square statistics for sex susceptibility for each of the disorders separately. It seems that a majority of the population do not perceive differences in sex susceptibility in relation to each of the disorders. Hence, it might also be useful to compare those who identified differences as opposed to those who did not, for each of the disorders. A similar analysis in relation to the other examinations done for sex susceptibility would be useful.
4) Lines 207-213- it would be helpful to provide a rationale for this examination. As the question regarding sex susceptibility is directly asking about which sex is more vulnerable to the disorder, there might be no association between the participant’s response and sex of the protagonist. Hence, this type of examination might be more relevant to an examination of methodological issues and how naming a particular sex in the vignette can affect participant responses.

Discussion
5) It would also be useful to discuss the findings in relation to need for treatment for the different disorders being different between the sexes.
6) 245-247- it is difficult to comment on the improvement of mental health literacy in relation to perceived seriousness as a comparison has not been provided
7) Line 265- 270- consider issues that were raised earlier that the results show that a majority of the sample perceive that there is no difference in sex susceptibility when the protagonist was either male or female.
8) Summarise the limitations section. Also consider the relevance of the limitations that have been discussed. For example, what is the relevance of the limitation provided in lines 288-292 to this study?
9) Describe the implications of the findings.

Additional comments

No further comments

---

## Round 0.2 · Minor Revisions

With the change of title, the theme of sex differences is not apparent to the reader until reading the abstract. I think this focus needs to be mentioned somewhere in the title.

Page 2. In the Objectives section of the Abstract, the phrase “the influence sex has” may initially confuse the reader about the sense that “sex” is being used. I suggest a rewording, such as “to examine sex differences in…”. An alternative would be to use the term “gender” throughout, which is probably more accurate.

It is worth mentioning in the Abstract that the sex of the vignette was varied as well, e.g. “were randomly assigned either a male or female version of a vignette…”.

On page 6, the phrase “perceived sex susceptibility” could be stated more clearly, e.g. “the perceived susceptibility of each sex to mental illness”.

Page 10 has a double comma.

---

## Round 0.3 · accepted · Accept

Thank you for making the suggested revisions.